# Ants Can Anticipate the Following Quantity in an Arithmetic Sequence

**DOI:** 10.3390/bs11020018

**Published:** 2021-01-28

**Authors:** Marie-Claire Cammaerts, Roger Cammaerts

**Affiliations:** 1Biology of Organisms Department, University of Brussels, 1050 Brussels, Belgium; 2Natural and Agricultural Environmental Studies Department (DEMNA) of the Walloon Region, 5030 Gembloux, Belgium; rogercammaerts@gmail.com

**Keywords:** anticipatory behavior, associative learning, episodic-like memory, forethinking, *Myrmica sabuleti*, operant conditioning

## Abstract

Workers of the ant *Myrmica sabuleti* have been previously shown to be able to add and subtract numbers of elements and to expect the time and location of the next food delivery. We wanted to know if they could anticipate the following quantity of elements present near their food when the number of these elements increases or decreases over time according to an arithmetic sequence. Two experiments were therefore carried out, one with an increasing sequence, the other with a decreasing sequence. Each experiment consisted of two steps, one for the ants to learn the numbers of elements successively present near their food, the other to test their choice when they were simultaneously in the presence of the numbers from a previously learned sequence and the following quantity. The ants anticipated the following quantity in each presented numerical sequence. This forethinking of the next quantity applies to numerosity, thus, to concrete items. This anticipatory behavior may be explained by associative learning and by the ants’ ability to memorize events and to estimate the elapsing time.

## 1. Introduction

Among the animal actions that prepare them for the future, there are genetically driven patterns such as seasonal behaviors that are controlled by environmental changes and hormones [1]. These fixed action patterns do not necessitate that the animal has a sense of the future or must learn to perform these patterns. Other future-directed behaviors may be a consequence of prospective thinking. Scrub jays, for example, can anticipate future needs in food in response to a learned situation and independently of their present needs [2,3]. This kind of planning for the future was considered by Shettleworth [4] to be genuine because it implements a novel action and is appropriate to a future motivational state distinct from that of the moment. This planning for the future leans on the jays’ episodic memory [5], a cognitive faculty enabling the recollection of past lived events with their sequential order. For this kind of memory, Clayton and Dickinson [5] used the term “episodic-like memory” since it is impossible to ask animals whether they consciously remember an event or whether they know what they experienced.

In mammals such as rats, the hippocampus plays an essential role in the memorization of previous odors together with their sequential order, although it is not required to recognize recent odors [6]. The crucial role of this organ in episodic memorization as well as in imagining the future is revealed in humans with brain damage limited to the hippocampus. Such patients are amnesic for episodic experiences and cannot imagine personal future episodes nor generate hypothetical scenarios. Moreover, their mental representation lacks spatial coherence [7]. Functional magnetic resonance imaging brain studies showed increased activity in a brain network, including the hippocampus, when people recalled episodic memory and imagined future episodes [8]. It thus appears that the same neural network enables reliving past events (the episodic memory) and projecting oneself into the future in a mental journey in time from the present moment. The latter capacity to organize current action in view of anticipated events has been called episodic foresight [9] or episodic future thinking [10]. The forethinking of possible future events may thus be based on the memorization of lived events and their position in time.

Not all animals are equally expert in planning for the future. An example is a difference between rats and squirrel monkeys as for their next choice of food. Deprived of water and given a choice between large or small amounts of food, only the monkeys, not the rats, chose small amounts of food, enabling later on to cause less thirst [11].

Another kind of planning for future needs is pre-experiencing an upcoming event and behaving accordingly. For example, great apes, among other behaviors, can save tools for future use [12]. They even show self-control in relation to delayed events and generalize a function from a novel object to a future use [13].

However, recent theoretical modeling [14] and simulations [15] show that behavioral decisions about a future action may also result from associative learning. Learning by chaining (i.e., linking together) sequences of behaviors through conditioned reinforcement can lead to making decisions about future states that lack immediate benefits. This kind of associative learning may explain the occurrence of anticipatory behaviors in great apes and even their flexible planning [15].

Vertebrates, especially birds and mammals, show the most derived numerical abilities (references in [16,17]). Some studies on the numerical skills of monkeys [18] and chimpanzees [19] suggest that (although this was not the object of these works), these primates would be able to anticipate the next quantity in a numerical sequence. We should also expect that the most behaviorally advanced invertebrates, i.e., the social Hymenoptera, present some future-oriented behaviors. The exploitation of flowers by bumblebees, for example, is governed by the expectation of a threshold volume of nectar under which the insect leaves the feeder plant [20].

Concerning honeybees, one should expect that they will be able to react anticipatively with regard to quantities because they have a suitable level of numerical skill. Indeed, they can discriminate (small) numerosities [21], can add and subtract numbers of elements [22,23], have the notion of zero [24] and can acquire numerosity symbolism [25].

Ants are not devoid of skills in anticipative behavior. Workers of *Myrmica sabuleti* Meinert, 1861 (Hymenoptera, Formicidae) by living in colonies maintained in the laboratory can anticipate the time as well as the location of food deliveries [26,27]. This kind of behavior is not unforeseen since workers of this ant possess several cognitive abilities. Among others, they can recognize themselves in a mirror though having probably no self-awareness, solve simple problems (e.g., walking around a barrier), learn to react to novel situations (e.g., pulling on a double door), learn a behavioral sequence, and can acquire serial recognition, but only if rewarded [28,29,30,31,32]. They can present an acquired conditional behavior in a subsequent situation (e.g., having been conditioned at the same time to a cue associated with meat and to another cue associated with sugar water, when deprived of meat, they react to the cue that was previously associated with meat, and when deprived of sugar water, they react to the one that was previously associated with sugar water [33]). They also possess several numerosity abilities. Among the latter, they natively have a left-to-right-oriented number line, can acquire the notion of zero through experiences, can add and subtract numbers of visual or of olfactory cues if seeing the result of the operation during training, can acquire symbolisms including a symbol for zero, and can use learned symbols for adding and subtracting [16,17].

Since the ant *M. sabuleti* has some cognitive and numerosity abilities and can anticipate the time and location of its next food delivery, it was logical to examine if it could also present numerosity anticipatory behavior. We examine here if *M. sabuleti* workers were capable of expecting a larger or a smaller number of elements than the last sighted in a sequence displayed over time near their food site when the sequence was increasing or decreasing. More precisely, we first trained them to successively 1, 2, 3, and 4 elements (experiment I) or to successively 5, 4, 3, and 2 elements (experiment II) set near their food versus 0 (i.e., the absence of an element). Note that these ants have the notion of zero [34] and correctly locate it at the start of an increasing and at the end of a decreasing series [35]). For the four training sessions to be each conditioned to a number of the progressive sequence of elements, it was experimentally checked if the ants acquired an association between the presented number and the food. Second (i.e., after the last training and checking sessions), the ants were tested simultaneously in front of the four numbers of elements to which they have been trained and to a fifth number, which was the next expected amount in the presented sequence. If for the increasing and for the decreasing sequence, this next number of the element(s) was preferably chosen, then it could be concluded that the ants present anticipatory behavior for numerosity in a sequence.

## 2. Material and Methods

### 2.1. Collection and Maintenance of Ants

The experiments were performed using two colonies of *M. sabuleti* collected in September 2019 at Olloy/Viroin (Ardenne, Belgium) in an abandoned quarry. These colonies each contained about 500 workers, a queen and brood, and were living under stones and in the grass. Each of the two colonies was maintained in the laboratory in one to two glass tubes half-filled with water with a cotton plug separating the ants from the water. The nest tubes of each colony were deposited in a polypropylene tray (30 cm × 15 cm × 5 cm) where the borders were lightly covered with talc to prevent escape. The trays served as foraging areas: pieces of *Tenebrio molitor* larvae (Linnaeus, 1758; Coleoptera, Tenebrionidae) were deposited three times per week on a piece of a microscope slide, and sugar water was permanently provided in cotton plugged tubes. The ambient laboratory temperature was ca 20 °C, the humidity ca 80%, the lighting 330 lux while working on ants, and the electromagnetism 2 µWm^2^. These environmental conditions were adequate for the species.

### 2.2. Experimental Planning

The first series of experiments using blue cues (=experiment I) aimed to determine if *M. sabuleti* ants could anticipate the next number of elements in an increasing sequence of numbers presented over time near their food (the reward). Using operant conditioning, the ants were first successively trained to 1, 2, 3, then 4 elements set near their food against zero elements (a blank stand) set far from food at days 1, 3, 5, 7, respectively. They were tested at days 2, 4, 6, 8 to check if they effectively learned the association between their food and the presented cues. In a second part of the experiment, on day 9, the ants were tested facing 1, 2, 3, 4 and 5 blue elements presented simultaneously all around them. If the ants mostly responded to 5 blue elements, then we should conclude that they behaved expecting the next quantity in the increasing sequence. Incidentally, they would also have made an addition by adding 1 to 4.

Five days later, the experiments were repeated using yellow cues. It should be noted that blue and yellow are two very distinct colors, very well perceived by *M. sabuleti* ants [36]. This second series of experiments (=experiment II) aimed to examine if the ants could anticipate the next number of elements in a decreasing sequence of numbers presented over time near their food (the reward). Using operant conditioning, the ants were successively trained to 5, 4, 3, then 2 yellow elements set near their food against zero elements (a blank stand) set far from food at days 1, 3, 5, 7, respectively. They were tested on days 2, 4, 6, 8 to check if they effectively learned the required association between the presented numbers of cues and the food. In a second experiment, on day 9, the ants were tested in front of 5, 4, 3, 2, and 1 yellow element presented simultaneously all around them. If the ants mostly responded to 1 yellow element, then we should conclude that they behaved expecting the next number in the decreasing sequence. Incidentally, they would also have made a subtraction by subtracting 1 from 2.

### 2.3. Experimental Materials

The cues that were presented to the ants were blue or yellow circles (diameter = 0.2 cm) drawn inside a white (blank) square (2 cm × 2 cm) using Microsoft Word^®^ software. The squares with 1, 2, 3, 4 or 5 blue or yellow squares were printed, cut, and tied with extra transparent sticky paper on the front face of a stand. The cues were printed and tied a fortnight before starting the experimental work; they had thus no particular odor. Each stand was made of Steinbach^®^ (Malmedy, Belgium) strong white paper (250 g/m^2^), had a vertical part (2 cm × 2 cm) and was maintained vertically thanks to a horizontal part [2 × (1 cm × 0.5 cm)] duly folded (see photos in Appendix A.

The ants were first trained in their foraging area (Figure 1A). In the course of this training, they were tested in another tray (21 cm × 15 cm × 7 cm, the borders of which having been lightly covered with talc) (Figure 1B). In a second experiment, on day 9, they were tested inside an enclosure made of strong white paper (Steinbach^®^), the dimensions of which being: height = 3 cm, diameter = 10 cm, perimeter = 31.4 cm with 2.2 cm more for seam allowance. The enclosure for each colony was set in a tray whose bottom had dimensions of 25 cm × 10 cm (Figure 1C). Cues like those used for the first part of experiments I and II were arranged along the rim of the enclosure.

### 2.4. Experimental Methods, Assessments, Statistics

During the ants’ training, for each colony and each number of circles presented, the ants present near the provided cues were counted six times each two days (number of counts for each presented number of circles = 2 days × 2 colonies × 6 times = 24). Such counts were made to check if ants could sufficiently and equally perceive the different numbers of circles for acquiring conditioning. The numbers of ants counted near the different cues were compared to those expected if the ants equally foraged during their four successive training using the χ^2^ nonparametric goodness-of-fit test. The means of these counts are given in Table 1 andTable 2.

During the first part of experiments I and II, in order to check the efficacy of their conditioning, the ants were tested after having been transported in a separate tray (Figure 1B). In this tray, the “correct” cue was randomly set on the left or right side. To perform a test using a colony, 25 ants were transferred into their tray devoted to testing in front of the two cues provided during the training session preceding the test. The ants perceived the cues and freely moved towards them. They stayed 2 to 20 s near those of their choice. The numbers of ants sighted at a distance less than 2 cm of each kind of cue were punctually counted 20 times (every 30 s) over 10 experimental minutes. After each test, the ants were returned to their foraging area near their nest entrance. The numbers obtained for the two colonies were added in their chronological order, and the 20 corresponding sums obtained were ordered in four, thus producing 5 groups of numbers. The 5 successive numbers thus obtained for the “correct” cue were compared to the 5 successive numbers obtained for the “wrong” cue (that set far from food during training) using the nonparametric matched-pairs signed-ranks test of Wilcoxon. The critical one-tailed *p* value was read in the table for small sample sizes given in [37].

The second part of experiments I and II aimed to assess the ants’ response to the numbers of elements to which they had been trained as well as to one more element or one less element than the last number of elements to which they were trained. To perform a test on a colony, 30 ants (instead of 25 since in the round testing area the ants could visit 5 cues while when testing their association between food and a cue, they could visit only 2 cues) were transported in the center of their enclosure devoted to testing. The ants sighted the five cues located in the enclosure and freely moved towards those of their choice. For each two used colonies, the ants approaching each of the five cues set in the enclosure were counted 20 times over 10 experimental minutes. For each cue, the numbers obtained for the two colonies were added, and the five numbers were compared to those expected if ants randomly visited the five presented cues using the nonparametric χ^2^ goodness-of-fit test.

## 3. Results

### 3.1. Anticipation of the Following Number in an Increasing Arithmetic Sequence

First part of the experiment

During their training, the ants were equally numerous all around the different cues bearing elements (χ^2^ = 0.36, df = 3, 0.90 < *p* < 0.95, the mean values are in Table 1, the first part of the experiment, column 1). They thus could equally see and memorize these cues. They are known to acquire conditioning within the here used time period [36,38]. The proportion of correct responses to the sighted numerosity was high. The ant’s score relative to the learning of 2 circles was lower than that relative to 1 circle; that relative to 3 circles (at day 6) was higher than that relative to 2 circles (at day 4), and that relative to 4 circles (at day 8) was the highest (Table 1, First part of the experiment, columns 4 and 5). The level of all these conditionings was significant (*p* = 0.031: Table 1, First part of the experiment, columns 6 and 7).

Second part of the experiment

Faced with 1 to 5 blue circles, the ants of the two colonies went mostly to 5 circles. Together, they responded to 5 circles with a score of 68.8%, while they responded to 1, 2, 3 and 4 circles with scores of, respectively, 7.1%, 3.6%, 8.0%, and 12.5%. The ants’ preference for the 5 circles was highly significant (*p* < 0.001: Table 1, Second part of the experiment). The ants have thus expected that the next number of circles present near their food should be 5 (see also the Discussion section).

### 3.2. Anticipation of the Following Number in a Decreasing Arithmetic Sequence

First part of the experiment

During their training, the ants were equally numerous all around the different cues bearing elements (χ^2^ = 0.09, df = 3, *p* > 0.99; the mean values are in Table 2, first part of the experiment, column 1). They could thus see these cues and memorize them, being effectively able to acquire conditioning within the allotted time [34]. During the testing, the proportion of ants’ correct responses to the sighted numerosity was high. The ant’s score relative to the learning of 4 circles was lower than that relative to the learning of 5 circles; those relative to 3 circles and to 2 circles (at days 6 and 8) were higher than that relative to 4 circles (at day 4) (Table 2, First part of the experiment, columns 4 and 5). Although different in their level, each of these conditionings was significant (each time: *p* = 0.031) (Table 2, First part of the experiment, columns 6 and 7).

Second part of the experiment

Faced with 1 to 5 yellow circles, the ants of the two colonies went mostly to 1 circle, this preference being highly significant (*p* < 0.001: Table 2, Second part of the experiment). Together, they responded to 1 circle with a score of 65.4%, while they responded to 2, 3, 4 and 5 circles with respective scores of 19.1%, 8.1%, 2.9%, and 4.4%. Consequently, the ants correctly expected that the number of circles present near their food should be 1 (see also the Discussion section).

We remark that the ants’ responses during the first part of experiments I and II were very similar (Table 1 and Table 2, upper chart; Figure 2 left graph). Moreover, as can be easily seen in Table 1 and Table 2 (lower charts) and in Figure 2 (right graph), the ants’ responses during the second part of the experiments I and II were of equivalent strength (χ^2^ = 2.62, df = 4, 0.50 < *p* < 0.70).

## 4. Discussion

Since the workers of the ant *M. sabuleti* possess many numerosity abilities and can anticipate the following location and time of food delivery, we examined if they could expect a larger or a smaller number of elements presented near their food when this number increased or decreased over time. Our results, graphically summarized in Figure 2, showed that the ants expected that the next number of elements of the sequence to which they were conditioned should be larger or smaller. They thus presented an anticipatory behavior relative to numerosity. The obtained ants’ conditioning scores (during the first part of the experiments) were in agreement with those commonly observed in this species, which generally equaled ca 80%, e.g., in [38,39,40]. In addition, the final scores of the experiment I (increasing sequence of numbers) were statistically equivalent to those of the experiment II (decreasing sequence of numbers).

The learning of the second presented cue (i.e., two elements in the increasing sequence and 4 elements in the decreasing sequence) led to a lower conditioning score (about 70%; Table 1 and Table 2 and Figure 2) than that of the first presented cue (respectively 1 and 5 elements). The explanation may be that the ants were then still conditioned to the first cue (ca 80% of correct responses). Thereafter, when the third and the fourth cues were presented, the ants perceived that they were in the presence of an increasing or decreasing sequence of items and their proportion of correct answers became higher. This was the most obvious for the increasing sequence, where the last cue elicited ca 90% of correct responses.

The fact that the ants chose the following quantity in a decreasing sequence as well as in an increasing sequence shows that the ants’ response was not directed towards a larger or a smaller visual stimulus but was really the anticipation of a future event on the basis of a past experience (see below what concerns episodic memory).

Incidentally, during the two experiments, the ants performed the addition 4 + 1 and the subtraction 2 − 1. This presumption of really making an arithmetic operation is in the course of being checked in a dedicated experimental work.

In order to anticipate an event such as a number of elements in an increasing or decreasing sequence, the ants should have memorized formerly experienced events and their order of occurrence. To do so, the ants must possess two abilities: evaluation of the running time and memorization of experienced events. They effectively do possess these two abilities: they evaluate the running time [41] and navigate using memorized cues [42]. Assuming that ants have an episodic-like memory, by what mechanism could they anticipate the future terms in a sequence of numbers?

Before answering this question, it should be noted that this and similar works must be performed on foragers, i.e., old and experienced workers. Indeed, these 2- to 3-year-old ants have the two required abilities reported above, i.e., memorizing experienced events and perceiving the running time. Young ants, on the contrary, can acquire conditioning (so can memorize elements and events), but they may not detain the notion of running time.

Planning future actions requires the mental ability to represent what will happen, where and when, as well as the flexible ability to act according to a future motivational state [43,44]. Here, the ants could only represent what may occur (a successive number), when (after the removal of the last number) and where (near their food). One cannot demonstrate that the experimented ants had the ability to mentally travel forward in time, but they have never been in need of food (which was continuously renewed as necessary), and consequently, they may have reacted according to a future and not a current motivational state. It is thus more likely that their behavior was not foresight (future-oriented) but that it relied on temporal updating of an associative learning process (see next paragraph). Up to now, future goal-oriented cognition in animals appears to have been unambiguously demonstrated only in great apes and in jays. Great apes show flexibility in future-directed behavior such as delayed behavior (showing self-control ability) [45], as novel behavior [13], as deferred exchanges of objects in order to obtain a future reward [46,47] and as taking cover for future deception [48]. More remarkable, wild orangutan males communicate vocally, in advance, their future travel direction [49]. As for jays, they use previous experience to modify their food-caching strategies [43].

However, an alternative to flexible planning behavior emerging from mental time travel with a sense of the future is associative learning, a simple mechanism that could explain anticipatory behavior [14]. Indeed, based on the results of experiments with great apes [12] and ravens [50], simulations have shown that through conditioned reinforcement, flexible planning behavior, and thus decisions about future states can simply emerge from associative learning [15].

The ants’ anticipation of the following quantity in a numerosity sequence might thus be simply derived from operant conditioning. Since when under conditioning, episodic memorization allows learning four successive counts, the ants could acquire a mental representation of the sequence. They could thus forethink the following numerosity in the sequence and select what will be the following cue simply because the cue was always set near the food. This process of associative learning may be one of temporal updating, which is maintaining information over time and updating it as new information becomes available [51] (p. 2), the learned successive temporal locations of the memorized items being represented within a mental temporal map [51] (p. 20). Moreover, each learned cue might reduce the response to the preceding one, a process known as blocking [52], so that it is the last conditional sighted cue that should attract the most of the ants. This kind of foresight does not require a mental journey through time with a sense of the future, a notion that would imply that of *autonoesis*, the awareness of one’s own existence as a temporal entity.

The artificial situations in which the ants were here trained may have some correspondence in their day-to-day life in the wild. Operant conditioning, which occurs in nature each time an individual is rewarded after having presented a given behavior, allows acquiring behavioral patterns optimally adapted to environmental circumstances [14]. It may be advantageous for foraging ants to remember not only the location of past encountered food sites but also the, over time, decrease or increase of food items and the changes in plant materials present on these sites. Episodic forethinking allows doing so. This ability produces future performance benefits such as decision-making and spatial navigation [10].

In bees, electrophysiological work has recently shown that a particular part of the brain is devoted to allowing rather complex numerosity abilities [53]. In ants, nothing is yet known about their brain functioning with respect to numerical abilities, memorization of events, memorization of sequences, and assessment of running time. Nevertheless, investigation on their brain functioning under chemical stimulation has already been conducted [54], as well as on neuromodulation in relation to their social behavior [55]. Electrophysiological studies similar to those conducted on bees should be undertaken on the ants’ numerosity ability.

It must be noted that forethinking behavior has always been observed in advanced animal species. The originality of our present work is that the revealed anticipatory behavior concerns the numerosity ability of an invertebrate. In ants, the ability to anticipate may not be innate but may likely be acquired through experiences. Indeed, only foragers and not yet young ants can expect the future location and time of food delivery [56]. The same could be true for the expectation of the following number in a sequence.

## 5. Conclusions

We demonstrate that *M. sabuleti* ants maintained in laboratory settled colonies could forethink what would be the following quantity in a short arithmetic sequence of a number of elements associated with their food. They behaved for numerosity as they did for the time and the location of the following food delivery. They detain this ability because they can estimate the running time and can memorize visual cues present in their environment and in the first place because they are able to acquire conditioning. Simple associative learning, together with the ants’ ability to memorize and evaluating the running time, could thus account for their anticipatory behavior. Their anticipation of the following quantity is not surprising since their numerosity ability is high, similar to that of vertebrates, although limited to a concrete level. In humans, anticipatory behaviors are more complex and reach an abstract level as they analyze situations and several parameters, plan what will happen, and assess the future consequences of different ways of acting. Anticipative behavior is probably not native in ants but most likely acquired in the course of their life. It should be interesting to investigate the potential anticipatory behavior presented by young ants as well as by other young animals, vertebrates or invertebrates such as bees, still without life experience.

## Figures and Tables

**Figure 1 behavsci-11-00018-f001:**
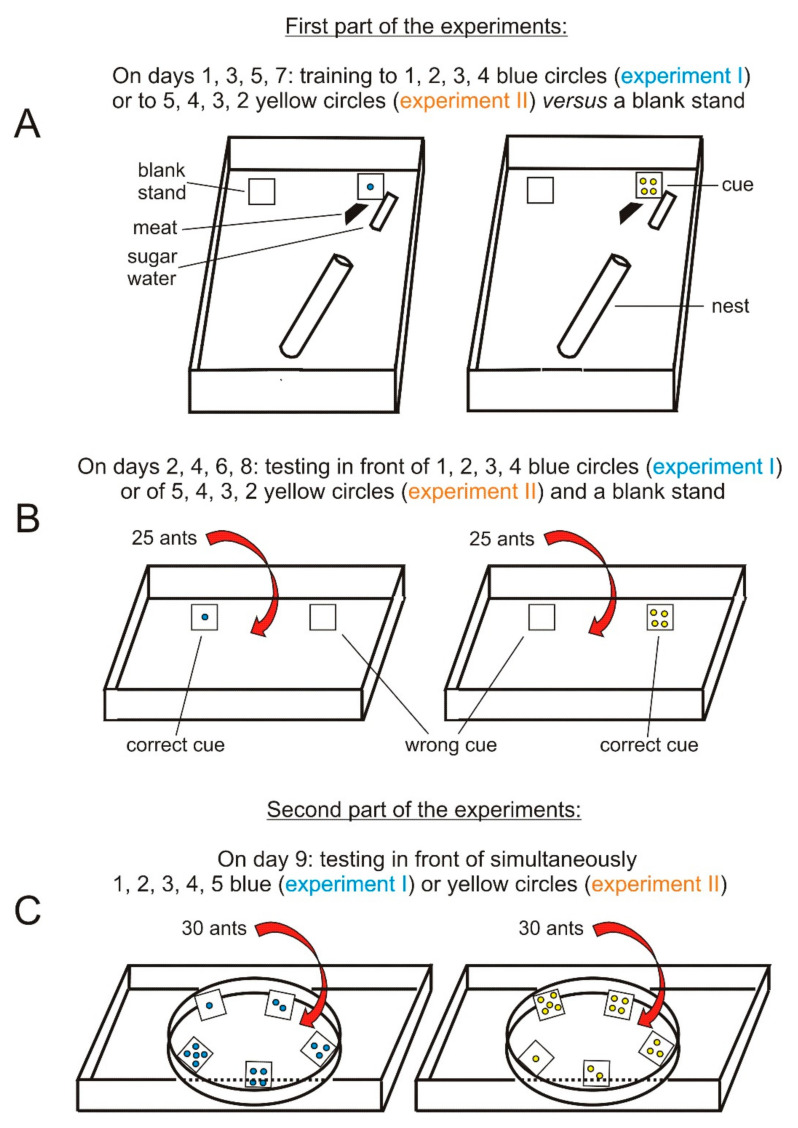
Experimental design. (**A**,**B**) concern the first part of the experiments: the ants of each of the two colonies were trained in their foraging areas to “correct” cues set near their food, which were successively 1 to 4 blue circles and thereafter 5 to 2 yellow circles, each of them versus 0 circles (a blank stand representing the “wrong” cue) set far from food. After each training using a number of circles, this conditioning was assessed by testing the ants in front of the “correct” and the wrong cues. (**C**) concerns the second part of the experiments: the ants were tested in front of 1, 2, 3, 4, and 5 circles placed along the rim of a circular enclosure.

**Figure 2 behavsci-11-00018-f002:**
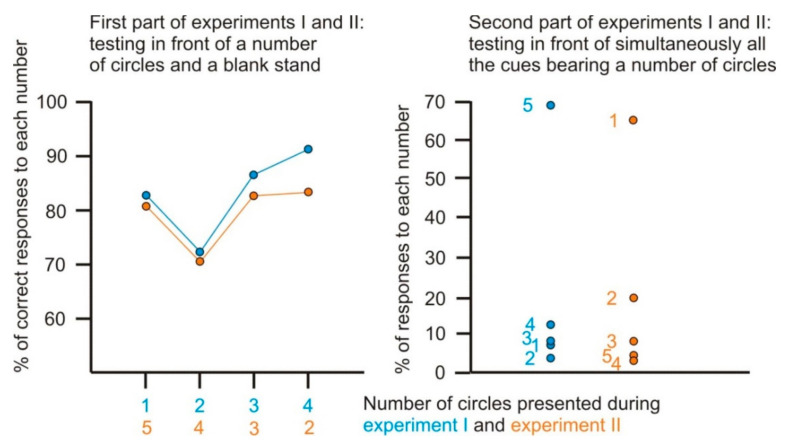
Graphical representation of the results. The ants of two colonies were trained to successively 1, 2, 3, 4 circles (blue dots) or 5, 4, 3, 2 circles (yellow dots) set near their food, and they duly acquired this learning (left graph). After this (right graph), they were tested in front of simultaneously 1, 2, 3, 4, 5 circles and reacted then mostly to the 5 circles (blue dots) or the 1 circle (yellow dots), having thus correctly anticipated the following number in an increasing and in a decreasing sequence of elements. A similarity appeared between the ants’ responses in the course of the experiment I relative to an increasing sequence (blue dots) and of the experiment II relative to a decreasing sequence (yellow dots).

**Table 1 behavsci-11-00018-t001:** Results of experiments made for examining if ants could anticipate the following term in an increasing sequence of numbers.

First Part of the Experiment.								
Training: Mean *n*° of Ants Near the Two Cues	Training: Number of Circles Borne by the Cues	Tests Made at Day	Tests: *n*° of Ants Sighted Near Each Cue, for Colony A; Colony B	Tests: % of Correct Responses	Tests: *n*°s of Ants of the Two Colonies Sighted Near Each Cue and Chronologically Ordered by Four			Wilcoxon Test N T P
4.3	1 vs. 0, at days 1, 2	2	64 vs. 7; 32 vs. 13	82.75	15,14,24,21,22 vs. 4,3,7,0,6			5 15 0.031
5.4	2 vs. 0, at days 3, 4	4	40 vs. 13; 51 vs. 22	72.22	17,16,15,21,22 vs. 9,7,6,5,8			5 15 0.031
6.0	3 vs. 0, at days 5, 6	6	38 vs. 8; 45 vs. 5	86.46	14,21,20,17,11 vs. 2,3,6,2,0			5 15 0.031
6.0	4 vs. 0, at days 7, 8	8	33 vs. 5; 51 vs. 3	91.30	13,16,19,18,18 vs. 1,0,3,3,1			5 15 0.031
**Second Part of the Experiment.**								
**At day 9, Presentation of 1, 2, 3, 4, and 5 Circles at the Same Time:**				***n*° of Ants Sighted in Front of Each Number:**				
				Colony A:	5,	1,	4,	11, 44
				Colony B:	3,	3,	5,	3, 33
				Total:	8,	4,	9,	14 77
				χ^2^ = 119,22, df = 4, *p* < 0.001				

The table gives the numerical results of the first part of the experiment, i.e., the ants’ training to 1, 2, 3, and 4 blue circles successively presented versus 0 circles and the ants’ testing in front of these cues, as well as of the second part, i.e., the ants’ testing faced with simultaneously 1, 2, 3, 4, and 5 blue circles. The ants acquired conditioning to the four successively presented cues and responded mostly to the following expected number (i.e., 5) of circles. They thus expected the following number in the arithmetic sequence. Schemas of the experimental design are given in Figure 1, photos in Appendix A and details in the text.

**Table 2 behavsci-11-00018-t002:** Results of experiments made for examining if ants could anticipate the following number in a decreasing sequence of numbers.

First Part of the Experiment								
Training: Mean *n*° of Ants Near the Two Cues	Training: Number of Circles Borne by the Cues	Tests made at day	Tests: *n*° of Ants Sighted Near Each Cue, for Colony A; Colony B	Tests: % of Correct Responses	Tests: *n*°s of Ants of the Two Colonies Sighted Near Each Cue and Chronologically Ordered by Four			Wilcoxon test N T P
5.5	5 vs. 0, at days 1, 2	2	68 vs. 15; 57 vs. 15	80.65	22,27,27,19,30 vs. 2,7,6,7,8			5 15 0.031
6.1	4 vs. 0, at days 3, 4	4	38 vs. 8; 55 vs. 31	70.45	12,20,23,22,16 vs. 5,7,11,8,8			5 15 0.031
6.1	3 vs. 0, at days 5, 6	6	43 vs. 12; 38 vs. 5	82.65	16,18,17,17,13 vs. 3,5,7,1,1			5 15 0.031
6.5	2 vs. 0, at days 7, 8	8	43 vs. 17; 61 vs. 4	83.20	21,20,21,23,19 vs. 1,4,4,8,4			5 15 0.031
**Second Part of the Experiment.**								
**At day 9, Presentation of 1, 2, 3, 4, and 5 Circles at the Same Time:**				***n*° of Ants Sighted in Front of Each Number:**				
				Colony A:	43,	17,	4,	3 5
				Colony B:	46,	9	7	1 1
				Total:	89,	26	11	4 6
				χ^2^ = 186.41, df = 4, *p* < 0.001				

The table gives the numerical results of the ants’ training to 5, 4, 3, and 2 yellow circles successively presented versus 0 circles, as well as of the ants’ testing in front of these cues (upper chart) and of the ants’ testing faced with simultaneously 1, 2, 3, 4, and 5 yellow circles (lower chart). The ants acquired conditioning to the four successively presented cues (upper chart) and responded mostly to the following expected number (i.e., 1) of circles (lower chart). This showed that ants could expect the following number in the arithmetic decreasing sequence. Schemas of the experimental design are given in Figure 1, photos in Appendix A, and details in the text.

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
