# Peer review of "Ants Can Anticipate the Following Quantity in an Arithmetic Sequence"

_behavsci, 2021, doi:10.3390/bs11020018_

Round 1

Reviewer 1 Report

The authors made extensive changes to the manuscript in line with my comments (and many of the comments from Reviewer 1). I consider the responses to my sugestions sufficient to make the manuscript acceptable. However, I noticed that one response to my serious methodological objection was not found in the revised manuscript. Authors should add the sentence: "The cues presented to the ants have been printed 15 days before starting the experiment. They have no odor.", as they have written in responces.
Best regards

Author Response

Response to referee 1

We thank the referee for having read and commented twice our paper. Of course, we have now added the sentence he required, in the ‘Experimental materials’ section (line 146, 147).

All other corrections are also written in red.

All these corrections largely improved our paper.

Marie-Claire and Roger Cammaerts

Reviewer 2 Report

In this manuscript, authors focused on very interesting aspect of animal behavior, anticipatory behavior relative to sequence of numbers. As ants are intelligent and have some cognitive abilities such as solving simple problems, learning behavioral sequences,  recognizing themselves in a mirror, authors simply investigated if ants in the laboratory are able to anticipate the future in a sequence of numbers. Ants were first trained with successive cues (1,2,3 and 4Blue/5,4,3 and 2yellow circles) next to their food.  The day after each training, ants where experimentally check to make sure that they have learned these cues. In the last test, ants were examined with a fifth number which was the next expected number in the sequence. Ants were not trained before with this fifth number in order to investigate their anticipatory behavior as for numerosity in a sequence.

This manuscript provides valuable knowledge about the amazing world of social insects, however,I do have a few suggestions and questions that I think the authors should address prior to publication.

Major concerns: 

  1. In these experiments, you tested all 30 individual ants in one arena. It's well known that ants have trail pheromone that impacts the behavior of another individual receiving it. So, If several ants gather in a specific spot, they can call and attract other ants to that  specific spot. This might affect your experiment and the results. I suggest investigating each ant individually to remove the impact of trail pheromone and other communication signals. 
  2. In these experiments, you selected yellow and blue color in the elements presented next to the food. Does color play a crucial role in associate learning and episodic-like memory? How does color affect ants learning and anticipating ability? It would be great to address these questions in the manuscript. 

 It is worth mentioning that not all species of ant posses blue receptor in their visionary system e.g highly evolved ants : Formica and Cataglyphis (Ogawa et al. 2015).  However, Myrmecia has the blue receptor.

Minore concern: 

Introduction, Line 40-42: This sentences sounds a bit unclear to me. You mentioned that hippocampus has a crucial role in episodic memorization in human and then you brough “ while neural and cognitive … “. This sentences starting with “while” is in the agreement with previous one or against it? I suggest to rephrase it. 

Introduction, Line 50-53: I found this sentence too long. I suggest breaking it to 2 sentences to make it easier to follow for readers.  

Material and Methods, Line 112 and 121: For the first time, you mentioned using color cues (Blue and yellow) in the experiment, but you postpone mentioning the reason for picking these 2 colors over the other ones to the line 134. In order to make it clear, please bring line 134 “It is known that ants perceive very well these colors [34].” Earlier in the text in the “Experimental planning” part and give more explanation for choosing these 2 colors. 

Experimental methods, Line 147: how many ants from each colony were trained? 25?

Experimental methods, Line 163-164: You wrote, the twenty numbers chronologically obtained for each cue were summed by four, which provided five groups of numbers. Why are these numbers sumed by 4? What do you mean with the 5th group of numbers? What does this 5th group of numbers present and stands for? 

Experimental methods: this part is not easy to follow this part of manuscript. It comes with so many questions and several times reading to understand how these calculations were done. However, the method part should be easy to understand for readers and repeatable for other researchers. Needs to be improved .

Table 1 and 2, Line 223- 234: These 2 tables and presenting numbers are confusing to me. It would be great to come up with a better way of presenting the result on the table and a better explanation. 

Author Response

Response to referee 2

We are very grateful to the referee for having made so many judicious comments on our manuscript. We duly made all the required corrections, which are (together with some corrections asked by other referees) in red in the text.

All the 191 corrections made or suggested by the referee were taken into account, with the following remarks:

Abstract: ‘elements’ are parts of a whole, the presented cue. Howard et al. [22, 25] use ‘adding or subtracting one element from a group of elements’, these elements being dots (as in Bortot [21]) or miscellaneous shapes (as in Howard). We thus prefer using the term ‘elements’.

Lines 41 and following ones: the sentence is now detailed and the lines 41-49 re-written, with the suppression of 2 references (former numbers [7] and [8]) and the adding of 4 new ones (new numbers [7-10], hence, from former number 9, a shift of +2 in the numbering of the references).

Line 55-58: the use of the term ‘ontogeny’ was superfluous and has thus been deleted.

Line 73 (actually 78-79): recognizing oneself in a mirror does not necessarily imply self-awareness, which is a more complex cognitive ability requiring knowing its origin, identity, social place and function, what is well or wrong doing, etc.

Line 77: “they can apply what they acquired through conditioning”: is re-written in actual lines 81-85. When ants are trained at the same time to a cue associated with meat and to another cue associated with sugar water, when deprived of meat, they react to the first cue, and when deprived of sugar water, they react to the other cue.

Actual lines 161-163, 200-202 and 217-219: a goodness of fit chi-square test is made and shows absolutely no significant difference between the numbers of ants having seen each of the successively cues bearing elements (numbers 1, 2, 3, 4 or 5, 4, 3, 2).

Lines 162-167 (actual lines 172-178) (Experimental methods): the way we used the Wilcoxon test is now better explained. The smallest number of pairs of observations necessary to lead to a possible significant result is N = 5 and was obtained by arranging the 20 numbers chronologically obtained by sums of four. The test ranks the differences between pairs of observations and takes also account of the direction of these differences. With N = 5, the maximum difference is ‘15’ (what occurred for each test), and this leads to P = 0.031. It explains why the statistical outcomes are here all the same (T = 15 and P = 0.031) although the raw data are not exactly the same.

Line 174 and following ones, actual lines 181-187: the procedure is now better explained.

Figure 1: has been split into sections A, B, C.

Tables 1 and 2: they have been largely improved, and are now even smaller.

The content of the previous lines 200-202 and 215-216 (the fact that the ants could expect a larger or a smaller number of circles) is now reported only in the Discussion, at lines 271-273.

Lines 290-292, actually 305-308: we hope having re-written this in a somewhat softer language.

Actual lines 317-319: our original sentence (lines 303-304) was correct: it concerns food-caching, the behavior of hiding food items for a later use, not food-catching.

Lines 331-333, actual lines 346-349: note that research that was made on the insects’ (bees and ants) brain concerns other cerebral faculties than those relative to numerosity and the running time notion.

Line 341, actual lines 354-357: the capability of ants to expect the following number in a sequence could be age-related: only the foragers, and not yet the young ants can expect the future location and time of food delivery [56].

We feel greatly indebted to referee 2 whose corrections and comments allowed us to largely improve our paper.

Marie-Claire and Roger Cammaerts  

Reviewer 3 Report

Please find my comments in the attached annotated manuscript.

Author Response

Response to referee 3

We are very grateful to the referee for having made judicious comments on our manuscript.

We here answer to his comments:

Major concerns:

  1. Myrmica ants deposit a trail only after having found food or a new nest site, and while returning to their nest. Under other circumstances, they do not trail.
  2. The kind of color does not affect the ants’ conditioning. If they see the color (and they do so, except for some red colors), they acquire condition as usually.

Minor concern:

Lines 43-50: this is now re-written with more details (lines 40-49); the sentences have re-written, with the suppression of 2 references (former numbers [7] and [8]) and the adding of 4 new ones (new numbers [7-10], hence, from former number [9], a shift of +2 in the numbering of the references).

Lines 54-59: have been re-written (lines 59-63) and, as required, a sentence has been broken into three ones.

Lines 112-121, actually 120, 130, 131: M. sabuleti can distinguish all the colors; blue and yellow are two very different colors, i.e. more distinct from one another than e.g. blue and green, green and yellow, blue and violet. Red is somewhat less perceived. We have added a small sentence in our text to define this.

Line 158-164, 200-202, 217-220: the number of ants present in the vicinity of the presented cues, although not precisely counted, was more or less always the same in the course of the experiment.

Lines 159-167 (actually 171-178): the way we used the Wilcoxon test is now better explained. The smallest number of pairs of observations necessary to lead to a possible significant result is N = 5 and was obtained by arranging the 20 numbers chronologically obtained by sums of four. The only possible outcome for P<0.05 is then 0.313. The test ranks the differences between pairs of observations and takes also account of the direction of these differences. It explains why the statistical outcomes are here are all the same (T = 15 and P = 0.031) although the raw data are not exactly the same. 

Tables 1 and 2: they have been improved to allow a better understanding of the experimental process and counting.

Marie-Claire and Roger Cammaerts  

Round 2

Reviewer 2 Report

Thank you very much for considering my comments and your revisions. This edition of your manuscript is greatly improved. My only remaining comment  is to reconsider tables in this manuscript. I believe it still has potential to be improved. Well done!

Reviewer 3 Report

Thank you very much for your thoughtful revisions. This edition of your manuscript is greatly improved over the previous. Grammatical restructuring and taking care to more adequately explain methodologies made for an excellent read. Besides the manuscript needing one more go-through to eliminate any remaining grammatical errors, my only recommendation is that you revisit table 1, as its formatting is still rather awkward. You might consider splitting it into two tables instead of using "upper" and "lower" sections of the table. Using a tabulated table instead of one using the insert-table function may also result in a cleaner looking product. Overall an excellent final product!